# The Face of Crisis: Examining Factors Affecting Nurses’ Professional Values During the COVID-19 Pandemic

**DOI:** 10.3390/nursrep15110388

**Published:** 2025-10-31

**Authors:** Withirong Sutthigoon, Manaporn Chatchumni, Ravinan Thatsiriniratkul, Nuchanard Kiennukul, Wanitcha Roungsri, Sasiwan Boonyatham, Pitchayapan Chantara

**Affiliations:** 1Nopparat Rajathanee Hospital, 679 Ram Inthra Rd, Bangkok 10230, Thailand; withirong1818@hotmail.com (W.S.); nuchanard1@gmail.com (N.K.); wanitcharoungsr@gmail.com (W.R.); s07pui@gmail.com (S.B.); keakae2003@gmail.com (P.C.); 2School of Nursing, Rangsit University, Pathum Thani 12000, Thailand

**Keywords:** job satisfaction, nurses’ professional values, organizational commitment, establishing objectives at work, nursing profession

## Abstract

**Background/Objectives:** Nurses’ professional values are closely linked to job satisfaction, organizational commitment, and establishing objectives at work. During the COVID-19 pandemic, these relationships became especially crucial, yet they remain underexplored in the context of Southeast Asia. This study aimed to examine how these organizational and motivational factors influence professional values among nurses in a large public hospital in Thailand. **Methods:** A cross-sectional survey was conducted with 336 registered nurses who had at least six months of experience caring for COVID-19 patients. Standardized instruments were employed to measure job satisfaction, organizational commitment, establishing objectives at work, and professional values, and correlation analyses were conducted to assess associations between variables. **Results:** Job satisfaction correlated positively with professional values (r = 0.505, β = 0.097), while organizational commitment showed the strongest association (r = 0.620, β = 0.528). Establishing objectives at work was also positively related, though to a lesser extent (r = 0.236, β = 0.086). **Conclusions:** These findings underscore the importance of supportive work environments that foster motivation, recognition, and career development in sustaining nurses’ professional values, especially in times of crisis. This study also serves as a reference for the development of the nursing workforce in post-pandemic healthcare systems, with implications for international health policy and workforce planning.

## 1. Introduction

COVID-19, caused by a single-stranded RNA virus of the *Coronaviridae* family, affects multiple organ systems, particularly the respiratory system. First identified in 2020, the spread of the virus rapidly escalated into a global health crisis [1]. Between March 2020 and June 2023, the World Health Organization reported approximately 767 million confirmed infections and 6.9 million deaths worldwide [2]. Healthcare professionals, especially nurses, have been disproportionately affected due to their frontline roles. Globally, an estimated 2.4 million healthcare workers were infected, resulting in 115,000 deaths, 6000 of which occurred in nurses [3]. In Thailand, from April 2021 to April 2022, there were 1,574,612 confirmed COVID-19 cases and 16,498 deaths. Among those infected were 880 healthcare professionals, among which 1 physician and 7 nurses lost their lives [4].

Beyond the physical risk, nurses faced significant psychological burdens, including burnout, emotional exhaustion, and mental health deterioration caused by prolonged exposure to the virus, increased workload, and social stigmatization [5,6]. These challenges threatened both the retention of nurses and the sustainability of professional values essential to delivering high-quality care [7].

Nursing professionalism is grounded in three key dimensions: care, professional role, and professional standards. “Care” involves advocating for patients while maintaining ethical and professional practice. The “professional role” encompasses the delivery of individualized care and collaboration in policy development. “Professional standards” refer to adherence to nursing quality benchmarks while acknowledging individual competency levels [7].

Although prior studies have explored job satisfaction and organizational commitment in nursing [8,9,10], they have typically examined these factors in isolation, without linking them directly to professional values or goal-setting behaviors. Moreover, few studies have explored these relationships in the context of a pandemic, particularly in Southeast Asia. This research gap is critical, as pandemic-related disruptions may influence how nurses align their goals and values with organizational expectations.

To address this research gap, this study investigates how job satisfaction, organizational commitment, and establishing objectives at work—with the latter defined as the process by which nurses set, prioritize, and align their professional goals with organizational standards—affect professional values. The study is grounded in Herzberg’s two-factor theory [11], which distinguishes between motivator factors (e.g., responsibility, growth, and recognition) that promote satisfaction and hygiene factors (e.g., working conditions, policies, and salary) that prevent dissatisfaction. This framework allows us to conceptualize how workplace conditions influence the core professional values of nursing, particularly under crisis conditions.

At a tertiary care hospital in eastern Bangkok, which served as the study setting, 28,130 outpatient COVID-19 cases, 2198 inpatient cases, and 287 deaths were recorded between 1 April 2021, and 31 December 2022. Among hospital staff, 329 were infected, including 114 nurses, representing 34.65% of the total infections [12]. These figures reflect the vulnerability of frontline nurses and highlight the urgency of developing strategies to protect and sustain the professional values that underpin high-quality care.

### 1.1. Research Framework

This study employs a theoretical framework grounded in Herzberg’s two-factor theory [11], which categorizes job-related factors into two groups: (1) Motivator factors (satisfiers), which include recognition, achievement, responsibility, personal growth, and meaningful work—elements that actively contribute to job satisfaction and intrinsic motivation. (2) Hygiene factors (dissatisfiers): These refer to external conditions such as salary, benefits, workplace policies, peer relationships, and job security, which do not directly motivate workers but are essential to preventing dissatisfaction.

Within this framework, organizational commitment represents a nurse’s emotional attachment and loyalty to their organization, while establishing objectives at work refers to the process of goal setting, strategic planning, and performance alignment. This process uses cognitive skills to identify, prioritize, and evaluate work goals, making it an essential behavioral component of professional performance [13].

Professional values in nursing, comprising care, professional role, and professional standards, represent outcomes influenced by these workplace factors. Figure 1 illustrates the hypothesized relationships between these variables, suggesting that improved job satisfaction, strong organizational commitment, and clarity in goal setting (establishing objectives at work) contribute to the reinforcement of professional values in clinical practice.

### 1.2. Aims

This study aimed to investigate the effects of job satisfaction, organizational commitment, and establishing objectives at work on nurses’ professional values. Furthermore, this study sought to identify key predictors of professional nursing values, including factors related to job satisfaction, organizational commitment, and establishing objectives at work, within the context of the COVID-19 pandemic.

## 2. Materials and Methods

### 2.1. Study Design

This study employed a cross-sectional descriptive design to examine how nurses’ professional values are influenced by job satisfaction, organizational commitment, and goal setting (establishing objectives at work). This design enabled the collection and analysis of data from the target population at a single point in time, allowing us to identify the relationships among the key variables. This approach was appropriate for evaluating the immediate effects of organizational and psychological factors on nursing personnel working in a high-pressure hospital environment during the COVID-19 pandemic. The study was conducted across multiple specialized inpatient units at Nopparat Rajathanee Hospital, Thailand, to enhance generalizability across diverse care settings.

### 2.2. Study Setting

This research was conducted at Nopparat Rajathanee Hospital, a large tertiary care facility under the Department of Medical Services, Ministry of Public Health, Thailand. The hospital has a capacity of 950 beds and employs 688 registered nurses, 121 nursing assistants, and 322 patient attendants. Specialized units involved in the study included Cohort 5/1, Cohort 5/2, the Intensive Care Unit (ICU), Airborne Infection Isolation Rooms (AIIRs), and other critical care and general service units.

### 2.3. Population Characteristics and Sampling

The target population comprised 336 registered nurses from the inpatient department who had provided direct care for COVID-19 patients for at least six months during the pandemic period (April 2021–April 2023). A purposive sampling technique was used to ensure that only nurses with relevant pandemic care experience were included.

Inclusion criteria: Registered nurses who had provided continuous direct care to COVID-19 patients for a minimum of six months.Exclusion criteria: Nurses in administrative or non-clinical roles, nursing students or trainees, and those who were on extended leave during the data collection period. Data collection occurred from December 2021 to April 2023.

### 2.4. Sample Size Determination

The sample size was determined using the recommended minimum of 30 cases per variable, as outlined by Hair et al. (2010) [14]. With four major constructions (job satisfaction, organizational commitment, goal setting, and professional values) comprising 11 measurable sub-variables, a minimum of 330 participants was required (30 × 11 = 330). The final sample size of 336 nurses slightly exceeded this requirement, ensuring statistical reliability.

### 2.5. Data Collection Instruments

Participants were administered a structured five-section questionnaire that integrated previously validated Thai-adapted tools and translated scales. It was organized as follows:Section 1—Personal Information: This section included demographic data such as age, gender, education, marital status, current role, specialized training, unit, COVID-19 care involvement, and work experience.Section 2—Job Satisfaction: This section included 42 items adapted from Herzberg’s theory, adapted to Thai by Wiriyakangsanon [15]. The six sub-dimensions included compensation and benefits (Items 1–8), professional autonomy (Items 9–13), nature of work (Items 14–22), organizational policy (Items 23–30), interpersonal relationships (Items 31–37), and professional status (Items 38–42), with the latter being the most prominent. Responses were given on a 5-point Likert scale (1 = strongly disagree to 5 = strongly agree).Section 3—Organizational Commitment: This section comprised 12 items based on the three-component model by Meyer and Allen [16], adapted to Thai by Wiriyakangsanon [15], and was scored on the same 5-point Likert scale.Section 4—Establishing Objectives at Work: This section comprised 10 items adapted from the Goal Adjustment Scale [17], measuring goal-setting and motivational behaviors.Section 5—Professional Values: This section comprised 28 items adapted from the Nurses Professional Values Scale-3 (NPVS-3) by Weis and Schank [7], covering three dimensions: care (Items 1–10), professional role (Items 11–20), and professional standards (Items 21–28).

All instruments used a 5-point Likert scale ranging from 1 (strongly disagree) to 5 (strongly agree). The results are presented as the mean and standard deviation to describe central tendency and variability.

### 2.6. Validation and Reliability

While some instruments had been previously translated to Thai, the research team conducted an updated evaluation of content validity and cultural appropriateness. A panel of three experts (one internal and two external nursing faculty members) assessed item relevance using the Index of Item–Objective Congruence (IOC): For job satisfaction, 41 of 42 items had IOC scores between 0.8 and 1.0; for organizational commitment, all 12 items had scores between 0.8 and 1.0; for establishing objectives at work, the scores of all items ranged from 0.4 to 1.0; and for professional values, all 28 items scored 1.0.

A pilot test with 30 nurses caring for COVID-19 patients was conducted. Internal consistency was measured using Cronbach’s alpha, and the scores were as follows: job satisfaction: α = 0.93; organizational commitment: α = 0.96; establishing objectives at work: α = 0.80; and professional values: α = 0.90. All of these values indicate acceptable to excellent reliability.

### 2.7. Ethical Considerations

Ethical approval was obtained from the Institutional Review Board (IRB) of Nopparat Rajathanee Hospital (Approval Nos. 11/2565 and 4/2566). All participants gave written informed consent after being informed of the study’s objectives and procedures, and participation was voluntary, with withdrawal permitted at any time without penalty. Confidentiality and anonymity were strictly maintained; no identifiable data were collected; and completed questionnaires were stored securely and destroyed after data analysis. The results were reported in aggregate.

### 2.8. Statistical Analysis

Data were analyzed using IBM SPSS Statistics for Windows, Version 28.0 (IBM Corp., Armonk, NY, USA). Descriptive statistics (frequencies, percentages, means, and standard deviations) were calculated for the demographic and main study variables. Pearson’s correlation coefficients were used to examine relationships between job satisfaction, organizational commitment, establishing objectives at work, and professional values. Multiple linear regression (entry method) was conducted to identify predictors of professional values. The independent variables were job satisfaction, organizational commitment, and establishing objectives at work, while the dependent variable was professional values. In addition to p-values, we reported standardized beta coefficients (β) and 95% confidence intervals (CIs) to indicate effect sizes and improve the interpretability of our findings.

## 3. Results

### 3.1. Participant Demographics

As shown in Table 1, the sample consisted predominantly of female nurses (94.6%) with an average age of 36.5 years (SD = 11.6). Most participants were single (64.6%), held a Bachelor of Nursing Science degree (94.0%), and worked as civil servants (97.3%). The average length of participants’ professional experience was 13.6 years (SD = 11.8), with nearly half (44.3%) having more than 11 years of experience. Participants were assigned to a range of units during the COVID-19 pandemic, including Cohort wards, the ICU, and the Emergency Room, reflecting varied exposure to high-acuity care settings.

### 3.2. Levels of Key Variables

#### 3.2.1. Job Satisfaction

Overall, nurses reported job satisfaction with a mean score of M = 3.57 (SD = 0.42). The highest satisfaction scores were related to professional status (M = 4.00) and interpersonal relationships (M = 3.98), suggesting strong perceptions of professional recognition and teamwork. Conversely, compensation (M = 3.16), organizational policy (M = 3.35), and job characteristics (M = 3.46) received lower scores, highlighting ongoing concerns about structural support and benefits (Table 2).

#### 3.2.2. Organizational Commitment

Organizational commitment had a mean score of M = 3.88 (SD = 0.48). The strongest commitment was toward achieving high-quality outcomes (M = 4.16) and applying professional knowledge effectively (M = 4.09), indicating a sense of ownership and responsibility for care quality among nurses during the pandemic (Table 2).

#### 3.2.3. Establishing Objectives at Work

Establishing objectives at work had a mean score of M = 3.41 (SD = 0.43). While nurses reported relatively high satisfaction in setting care goals for both COVID-19 and non-COVID-19 patients (M = 3.79 and M = 3.73, respectively), lower scores were observed in areas such as pursuing new goals (M = 3.57) and maintaining ongoing goal focus (M = 3.56), potentially reflecting workload strain (Table 2).

#### 3.2.4. Professional Values (Core Values of Nursing)

Overall, professional nursing values had a mean score of M = 4.10 (SD = 0.39). Among the three dimensions, caregiving (M = 4.39) received the highest rating, followed by professional standards (M = 4.09) and professional role (M = 3.97). These findings reflect the participating nurses’ strong adherence to ethical and patient-centered care, even in a crisis (Table 2).

### 3.3. Correlation Analysis

The Pearson correlation analysis (Table 3) revealed statistically significant positive relationships between all three predictors and professional nursing values. Organizational commitment showed the strongest correlation with professional values (r = 0.620, *p* < 0.01), followed by job satisfaction (r = 0.505, *p* < 0.01) and establishing objectives at work (r = 0.236, *p* < 0.01). Within the subscale of job satisfaction, professional status (r = 0.587) and interpersonal relationships (r = 0.508) showed the highest correlations with professional values [18], indicating that recognition and collegial support are key contributors to maintaining a professional identity (Table 3).

### 3.4. Regression Analysis

Multiple linear regression was conducted to determine the extent to which job satisfaction, organizational commitment, and establishing objectives at work predict professional nursing values. The model result was statistically significant (F (3, 332) = 72.723, *p* < 0.001) and explained 59.7% of the variance in professional values (R^2^ = 0.597) (Table 4).

#### 3.4.1. Regression Coefficient Interpretation

As shown in Table 5, two independent variables were significant predictors of professional values, while one did not reach statistical significance:Organizational commitment had the strongest standardized effect (β = 0.528, *p* < 0.01), indicating that it is the most influential factor.Job satisfaction had a smaller but significant effect (β = 0.097, *p* < 0.05).However, establishing objectives at work (X_3_) did not show a statistically significant predictive effect (β = 0.086, *p* = 0.05), suggesting that goal-setting behaviors may have only an indirect or limited relationship with professional nursing values.

These results confirm that enhancing organizational commitment and job satisfaction can lead to stronger professional values among nurses, whereas goal-setting alone may not directly predict those values. Multicollinearity diagnostics showed no violations (all VIFs < 2.3).

#### 3.4.2. Regression Equation:

Unstandardized form: Y = 0.090X_1_ + 0.432X_2_ + 0.079X_3_ + constant;Standardized form: Z = 0.097Z_1_ + 0.528Z_2_ + 0.086Z_3_.

In this equation, X_1_ represents job satisfaction, X_2_ represents organizational commitment, and X_3_ represents establishing objectives at work.

### 3.5. Summary of Findings

These findings demonstrate that organizational commitment and job satisfaction positively influence nurses’ professional values, with organizational commitment emerging as the most powerful predictor. Establishing objectives at work, while conceptually important, did not show a statistically significant direct effect, highlighting that goal clarity alone may not be sufficient to sustain nursing professionalism during public health emergencies.

## 4. Discussion

This study revealed a moderately positive correlation (r = 0.505) between overall job satisfaction and professional nursing values, consistent with the interpretation scale of Schober et al. [18]; however, specific job satisfaction components, such as compensation and benefits, job characteristics, and organizational policies, demonstrated only weak correlations (r < 0.40). These findings suggest that inadequate compensation or benefits may adversely affect service quality, which aligns with the study by Wiriyakangsanon [15], who reported moderate job satisfaction among nurses and highlighted dissatisfaction with compensation and benefits. Similarly, Sangprapai [19] observed that fair compensation and recognition during the early stages of the COVID-19 pandemic contributed to improved direct nursing care. In addition, Su et al. [20] emphasized that static organizational policies often failed to adapt to dynamic work conditions during the pandemic. International studies further support these observations, indicating that inadequate compensation and limited benefits are among the most critical factors contributing to global nurse turnover, particularly during health crises [21,22]. Collectively, these findings underscore the importance of effective resource allocation and responsive organizational policies to foster professional nursing values and enhance workforce retention.

Satisfaction with professional autonomy also correlated positively with professional values. During the COVID-19 pandemic, nurses frequently encountered restrictions to their autonomy due to stringent clinical protocols, which affected their perceived professional agency. Consistent with this observation, Wiriyakangsanon [15] found that autonomy was positively associated with nursing performance. At an international level, multi-country studies across Europe have reported that diminished decision-making autonomy during the pandemic was significantly associated with increased burnout and erosion of professional values among nurses [23,24]. These results reinforce the motivational aspect of Herzberg’s two-factor theory [11], in which autonomy and recognition function as motivators that strengthen professional commitment.

Interpersonal relationships within the workplace also played a crucial role in nursing performance. Nurses who maintained positive relationships with their colleagues and supervisors demonstrated enhanced collaboration and problem-solving capacity, ultimately improving patient outcomes. These findings support those of Srisom [25], who confirmed that effective teamwork contributes to job satisfaction and staff retention. Boonsanan [26] also found that teamwork significantly predicts organizational commitment, and that satisfaction with professional status was moderately correlated with professional values. Nurses who perceived themselves as respected contributors during the pandemic were more likely to uphold the professional standards of nursing. Mustikawati and Ernawaty [27] found that recognition of professional status was correlated with longer job tenure and enhanced quality of life. Similarly, studies conducted in Canada and Australia report that teamwork not only increases job satisfaction but also reduces staff turnover, thereby improving both patient safety and nurse retention [28,29].

Organizational commitment was identified as a strong correlation with professional nursing values (r = 0.620, β = 0.528) [18]. Nurses who exhibited high commitment applied their knowledge and skills to meet both patient care and organizational objectives, thus reinforcing their professional values. Recognition from peers, supervisors, and patients further strengthened pride and intrinsic motivation, which is consistent with findings by Puchkanit et al. [30], who reported that motivation, teamwork, and participation in quality-improvement initiatives significantly predicted nursing performance. Notably, the multiple regression analysis indicated that job satisfaction, organizational commitment, and establishing objectives at work collectively explained 59.7% of the variance in professional nursing values, highlighting their combined significance. International studies support these findings, with Su et al. [20] and Hanum et al. [31] reporting that strong organizational commitment among nurses was associated with greater intent to remain in the profession during crises. Therefore, reinforcing organizational commitment represents a globally relevant strategy to maintain professional values and workforce stability.

Although establishing objectives at work showed only a weak and non-significant predictive effect (β = 0.086, *p* = 0.050), even though its correlation with professional values was statistically positive (r = 0.236). This indicates that while goal-setting behaviors are associated with professional values, they do not directly predict them in this dataset. The weak correlation likely reflects contextual limitations during the pandemic—such as frequent shift changes, workload fluctuations, and uncertainty—which reduced the clarity of individual goals. Nevertheless, consistent with Locke and Latham’s [13] goal-setting theory, structured and meaningful objectives remain important for sustaining long-term motivation. Ekkapan et al. [32] also found that professional value alignment contributes to improved care quality. Hence, goal-setting should be supported institutionally as part of professional growth rather than viewed as a predictive determinant.

Work experience also contributed positively to professional values. Most participants had more than 11 years of nursing experience (Mean = 13.6 years), reflecting enduring professional commitment. Urairak [33] found that job satisfaction and professional development significantly influenced career commitment. Therefore, experienced nurses represent a critical workforce asset, and the provision of career advancement opportunities is essential to sustaining their engagement.

Furthermore, achieving meaningful objectives in the workplace was found to enhance nurses’ sense of professional accomplishment. Integrating disease-specific knowledge, fostering interdisciplinary collaboration, and ensuring structured discharge planning emerged as key elements in achieving this goal. Li et al. [34] observed that stress associated with disease-specific planning could negatively affect nursing performance; however, supportive organizational interventions may mitigate these effects. Our study reinforces that organizational support measures, such as continuing education and professional development programs, are vital to sustaining professional nursing values and retaining skilled personnel in the long term [35,36,37].

## 5. Limitations and Strengths

This study has several limitations that warrant careful consideration. First, purposive sampling was employed, which may have introduced selection bias and limited the representativeness of the sample to the broader nursing population. The use of such non-random selection methods reduces the ability to generalize findings beyond the specific context studied [1]. Second, although the sample size was adequate for statistical analysis, the study was conducted at a single tertiary hospital in Bangkok, which may limit the generalizability of our results to other hospitals, regions, or healthcare systems, particularly in rural or international contexts [2].

Third, the cross-sectional design of the study provides only a snapshot of associations at a single point in time, which limits our ability to draw causal inferences between job satisfaction, organizational commitment, goal setting, and professional values. Longitudinal or multi-site designs would be better suited to assess temporal relationships and causal pathways [3,4]. Fourth, the use of self-report questionnaires may be subject to social desirability bias and response interpretation variability, despite the efforts made to use validated instruments. Participants might have over-reported positive behaviors or under-reported negative experiences, which could affect the reliability of the responses [5].

Despite these limitations, this study possesses several notable strengths. The timing of data collection during the COVID-19 pandemic offers unique insights into how extreme stressors and rapid organizational change impact the professional values of nurses. Additionally, most of the participants had more than 11 years of experience (mean = 13.6 years), reflecting a seasoned workforce whose perceptions offer valuable perspectives on workforce resilience. Furthermore, the use of highly reliable validated instruments enabled a robust examination of key predictors such as professional autonomy, goal setting, and organizational commitment, which are critical in supporting and sustaining professional values in nursing [6,7]. These findings contribute to the evidence base for designing interventions that promote workforce retention, job satisfaction, and long-term professional growth, particularly in high-stress or rapidly evolving healthcare environments.

## 6. Conclusions

Throughout the COVID-19 pandemic, nurses have served as the backbone of healthcare systems, demonstrating extraordinary dedication, resilience, and professional integrity under pressure. Sustaining and strengthening these professional values, particularly in times of crisis, require deliberate institutional support. The present findings indicate that job satisfaction, organizational commitment, and establishing objectives at work play critical roles in predicting the level of professional values among nurses.

To uphold and reinforce these values, healthcare institutions must implement proactive measures such as establishing clear and adaptive organizational policies, especially in response to crisis conditions; ensuring fair compensation and benefit structures; and cultivating supportive work environments that encourage professional development and organizational loyalty. Prior studies have emphasized that organizational commitment is a key determinant of nurse retention and the delivery of quality care, particularly during public health emergencies.

Moreover, investment in continuous professional education is vital. Educational opportunities not only enhance nurses’ clinical competence and autonomy but also contribute significantly to job satisfaction and a sustained commitment to the profession. Structured career advancement, skill development programs, and recognition systems can empower nurses to uphold high standards of care and reduce nurse burnout and foster their long-term engagement with the profession.

By addressing these critical organizational and professional factors, hospitals and health systems can improve nurse retention, elevate care quality, and ensure that nurses remain motivated and well-prepared to uphold their professional values in both routine practice and times of public health crisis.

## Figures and Tables

**Figure 1 nursrep-15-00388-f001:**
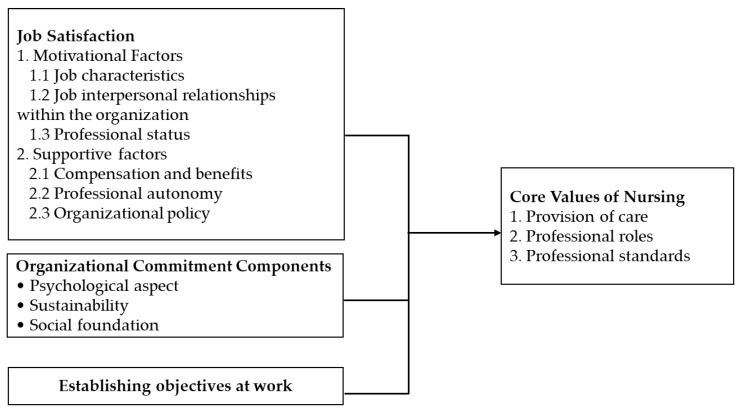
Research framework illustrating the hypothesized relationships, where job satisfaction, organizational commitment, and establishing objectives at work (independent variables) influence the core professional values of nursing care, professional role, and professional standards (dependent variables).

**Table 1 nursrep-15-00388-t001:** Participant demographics and work unit assignments (n = 336).

Variables		*N*	%
Sex			
	Female	318	94.6
	Male	18	5.4
Age (year)	Mean = 36.5, SD = 11.6, Min = 21, Max = 59		
	20–25	61	18.2
	26–30	90	26.8
	31–35	45	13.4
	36–40	24	7.1
	41–45	23	6.8
	46–50	32	9.5
	51–55	21	6.3
	56–60	40	11.9
Marital status			
	Single	217	64.6
	Married	105	31.3
	Divorced	14	4.2
Education level			
	Bachelor of Nursing Science	316	94
	Master of Nursing Science	14	4.2
	Master’s degree in another field	6	1.8
Position			
	Government officer	327	97.3
	Government employee of the Ministry of Public health	2	0.6
	Ministry of Public Health employees	7	2.1
Years of professional nursing experience (year): Mean = 13.6, SD = 11.8, Min = 1, Max = 38			
	1	19	5.7
	2–3	68	20.2
	4–5	36	10.7
	6–10	64	19
	more than 11	149	44.3
Locations where the sample group cared for COVID-19 patients			
	Cohort 5/1	46	13
	Cohort 5/2	57	16.9
	ICU Cohort and AIIR	33	8.8
	Cohort 5/1, ICU Cohort, and AIIR	45	13.4
	Cohort 5/1 and Cohort 5/2	55	16.4
	Cohort 5/2, ICU Cohort, and AIIR	15	4.5
	PICU	6	1.8
	Cohort ER	52	15.5
	Chemotherapy room	5	1.5
	NICU	4	1.2
	Operating room	11	3.3
	Special wards including internal medicine	7	1.9
	Single special ward for internal medicine	6	1.8

Legend: ICU = Intensive Care Unit; AIIR = Airborne Infection Isolation Room; ER = Emergency Room; PICU = Pediatric Intensive Care Unit; NICU = Neonatal Intensive Care Unit.

**Table 2 nursrep-15-00388-t002:** Mean, standard deviation, and interpretation of Job satisfaction, organizational commitment, establishing objectives at work, and core values of nursing (n = 336).

Items	$\bar{X}$	S.D.
Job Satisfaction		
Compensation and benefits	3.16	0.66
Professional autonomy	3.81	0.53
Job characteristics	3.46	0.49
Organizational policies	3.35	0.61
Interpersonal relationships	3.98	0.43
Professional status	4.00	0.47
Overall job satisfaction	3.57	0.42
Organizational Commitment		
Commitment to high-quality outcomes	4.16	0.57
Commitment to applying knowledge and skills successfully	4.09	0.47
Pride in receiving recognition	4.01	0.64
Commitment to the overall organization	3.88	0.48
Establishing Objectives at Work		
Caring for COVID-19 patients	3.79	0.61
Pursuing non-COVID-19-related goals	3.73	0.67
Pursuing new goals	3.57	0.79
Commitment to ongoing COVID-19 care goals	3.56	0.77
Establishing general objectives for work	3.41	0.43
Core Values of Nursing		
Caregiving	4.39	0.47
Professional role	3.97	0.46
Professional standards	4.09	0.43
Overall core values of nursing	4.15	0.39

*Legend:* All scales use a 5-point Likert rating (1 = strongly disagree, 5 = strongly agree). Possible score ranges—job satisfaction: 42–210 (subscale ranges: compensation and benefits: 8–40; professional autonomy: 5–25; nature of work: 9–45; organizational policy: 8–40; interpersonal relationships: 7–35; professional status: 5–25); organizational commitment: 12–60; establishing objectives at work: 10–50; professional values: 28–140 (subscale ranges: care: 10–50; professional role: 10–50; professional standards: 8–40).

**Table 3 nursrep-15-00388-t003:** Pearson correlation coefficients between predictors and professional values (n = 336).

	Core Values of Nursing	
	r	*p*-Value
Job Satisfaction by aspect		
Compensation and benefits	0.267 **	0.000
Professional autonomy	0.444 **	0.000
Job characteristics	0.355 **	0.000
Organizational policies	0.394 **	0.000
Interpersonal relationships	0.508 **	0.000
Professional status	0.587 **	0.000
Factors tested		
Job satisfaction	0.505 **	0.000
Organizational commitment	0.620 **	0.000
Establishing objectives at work	0.236 **	0.000

** *p* < 0.01.

**Table 4 nursrep-15-00388-t004:** Summary of regression model (ANOVA).

Source of Variance	Sum of Squares	df	Mean Square	F	Sig.
Regression	20.534	3	6.845	72.723	0.000
Residual	31.248	332	0.094		
Total	51.782	335			

**Table 5 nursrep-15-00388-t005:** Coefficients of multiple regression and diagnostics.

Variable	B	S.E.	Beta	*t*	*p*	Zero-Order	Tolerance	VIF
X_1_	0.090	0.058	0.097	1.542	0.024 *	0.505	0.457	2.188
X_2_	0.432	0.052	0.528	8.354	0.000 **	0.620	0.456	2.195
X_3_	0.079	0.040	0.086	1.955	0.050	0.236	0.935	1.070

The variables are defined as follows: X_1_ = job satisfaction; X_2_ = organizational commitment; X_3_ = establishing objectives at work. The symbols indicate the significance levels of the *p*-values: ** *p* < 0.01, * *p* < 0.05.

## Data Availability

The datasets generated and analyzed during this study are not publicly available due to ethical restrictions involving participant confidentiality. However, de-identified data may be made available by the corresponding author upon reasonable request and subject to approval by the appropriate institutional review board.

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
