# Peer review of "The Face of Crisis: Examining Factors Affecting Nurses’ Professional Values During the COVID-19 Pandemic"

_nursrep, 2025, doi:10.3390/nursrep15110388_

Round 1

Reviewer 1 Report

Comments and Suggestions for Authors

Hello, 

This is well done research article with great implications for healthcare systems. Thank you for it. Here are my suggestions.  

Line 35 – Remove “the” and “pandemic”. You are talking about Covid-19 itself. You introduce the pandemic later at lines 38 to 40. Therefore, I do not feel it is necessary in the first sentence. 

Line 42 – Remove “statistics reveal that out of the global figures”. I would simply state, “2.4 million health care workers were infected with Covid 19, resulting in...” 

Line 43 – Remove the word “that” after 6,000. 

Line 45 – I would add the word “total” after 1.5 million. The prior sentence talks about health care workers contracting Covid, but then you move into the total number of cases in Thailand, so I would specify to be clear.  

Line 46 – Among 880 health care professionals, only one doctor and seven nurses contracted Covid?? I just want ot make sure this statistic is clear.  

Line 48-50 – Is there a reference you can add to this? 

Line 57 and 59  – By whom?? Who is this by? It seems that the hero in white attire is a common phrase that is used? 

Line 70 – Grammar correction at the beginning of the sentence.  

Line 76 – Once again, by whom? 

Line 78 – You speak about Establishing Objectives at Work later in the paper and it makes it more clear, but it would be nice to see it explained more in this intro/background section. It’s in all capitals here but I have no idea what that is or the purpose of it while reading that sentence.  

Introduction in general – I struggled with determining “what is the problem”. Are the findings of the study to provide insight for healthcare organizations on how to create and implement strategies that motivate nurses to stay within the the system and to consistently uphold high standards after covid? Or during a pandemic? I was confused on the background. The introduction is what will capture your audience after the abstract, and I think that this could be reworked a little bit. The rest of the paper is very strong, however, I would like a little bit more details on what the purpose is in the introduction. 

Line 98 – Change font and size for reference.  

Like 104-105 – Capitalize E and W to stay consistent.  

Line 107 – Remove the word therefore.  

Line 107-111 - You don't talk about Covid 19 at all in the aims, but this is part of the title of the paper. 

Line 124 – For Figure 1, should the arrow be pointed the other direction towards satisfaction, commitment and objectives? I may be incorrect in my understanding of this.  

Line 143 – You describe EOAW here!!  

Line 144 – Reword to remove the word “us”. 

Line 257 – Table 1 demographics: Capitalize male, female, divorced, government officer.  

Line 362 – GREAT discussion section! 

Line 454 – This is repeated from the beginning of discussion section (line 415-416). 

Line 496 – Is this the first mention of continuing education? I didn’t see this anywhere else in the paper.  

Thank you!

Author Response

Comments

Revisions

Reviewer 1:  Recommendation: Revisions Required

Comments  1

Line 35 – Remove “the” and “pandemic”. You are talking about Covid-19 itself. You introduce the pandemic later at lines 38 to 40. Therefore, I do not feel it is necessary in the first sentence. 

We appreciate the reviewer’s suggestion. We have removed the words “the” and “pandemic” to focus the opening sentence directly on COVID-19 as the disease entity. The term “pandemic” is now introduced later in the paragraph.

Revised text (Lines 35–36):
“COVID-19, caused by a single-stranded RNA virus of the Coronaviridae family, affects multiple organ systems, with the respiratory system being the most significantly impaired.”

Line 42 – Remove “statistics reveal that out of the global figures”. I would simply state, “2.4 million health care workers were infected with Covid 19, resulting in...” 

Thank you for this helpful recommendation. We have simplified the sentence by removing the phrase “statistics reveal that out of the global figures” for clarity and conciseness.

Revised text (Line 41-42):
“Approximately 2.4 million healthcare workers were infected with COVID-19, resulting in 115,000 fatalities.”

Line 43 – Remove the word “that” after 6,000. 

We agree with this correction and have removed the word “that” to improve grammar and sentence flow. Revised text (Line 42-43):

“This figure includes a notably high number of nurses—6,000 of whom succumbed to the disease.”

Line 45 – I would add the word “total” after 1.5 million. The prior sentence talks about health care workers contracting Covid, but then you move into the total number of cases in Thailand, so I would specify to be clear.  

We agree that clarification is needed when shifting to national data. We have added the word “total” to specify that the figure refers to the overall number of COVID-19 cases in Thailand.

Revised text (Line 43-45):
“In Thailand, the situation reflected a similarly tragic pattern, with a total of 1,574,612 confirmed cases and 16,498 deaths reported between April 2021 and April 2022.”

Line 46 – Among 880 health care professionals, only one doctor and seven nurses contracted Covid?? I just want to make sure this statistic is clear.  

Thank you for pointing out the need for clarification. We have revised the sentence to clarify that among the total confirmed cases and deaths in Thailand, 880 were healthcare professionals infected with COVID-19. Of those, one physician and seven nurses died. Revised text (Lines 45–47):

“Among these cases, 880 were healthcare professionals who contracted COVID-19, including one physician and seven nurses who lost their lives—underscoring the risks faced by those on the front lines.”

Line 48-50 – Is there a reference you can add to this? 

We appreciate this helpful suggestion. To support the statement regarding the psychological challenges faced by healthcare workers during the COVID-19 pandemic, including burnout and emotional exhaustion—we have added two references:

  • Pappa et al. (2020), which conducted a systematic review and meta-analysis on depression, anxiety, and insomnia among healthcare professionals, and
  • Shanafelt et al. (2020), which discusses stress and anxiety among healthcare workers in high-pressure environments.

Revised text (Lines 50–53):

“Healthcare workers face not only the physical risk of infection but also significant psychological challenges. Extended working hours and the overwhelming influx of patients contribute to physical fatigue and emotional exhaustion, leading to conditions such as burnout and psychological distress [5,6].”

Line 57 and 59 – By whom?? Who is this by? It seems that the hero in white attire is a common phrase that is used? 

Thank you for highlighting the need for clarity. To avoid ambiguity, we have revised the sentence to eliminate the phrases “The Nurse Hero” and “hero in white attire.” Instead, we emphasize society’s recognition of nurses’ contributions and their professional dedication during the pandemic, which reflects a more academically appropriate tone.

Revised text (Lines 59–62):

“Consequently, society has acknowledged nurses as frontline professionals whose dedication underscores the profound value and heroism inherent in the nursing profession [7]. Nursing professionalism is conceptualized through three dimensions: care, professional role, and professional standards.”

Line 70 – Grammar correction at the beginning of the sentence.  

We have revised the sentence at line 70 to correct the grammatical structure. The revised sentence now reads more clearly and professionally.

Revised text (Line 84):

“One tertiary care facility with 590 beds located in eastern Bangkok reported handling 28,130 outpatient COVID-19 cases, 2,198 inpatient cases, and 287 deaths from April 1, 2021, to December 31, 2022.”

Line 76 – Once again, by whom? 

Thank you for the comment. We have clarified the source of recognition by indicating that acknowledgment came from society, the media, and professional institutions. This revision eliminates ambiguity and aligns with the academic tone of the manuscript.

Revised text (Line 89-92):

“Their exemplary performance during these challenging times has further reinforced societal and professional recognition of nurses as essential frontline professionals, highlighting their dedication and resilience.”

Line 78 – You speak about Establishing Objectives at Work later in the paper and it makes it more clear, but it would be nice to see it explained more in this intro/background section. It’s in all capitals here but I have no idea what that is or the purpose of it while reading that sentence.  

We appreciate the reviewer’s observation regarding the early mention of Establishing Objectives at Work in the Introduction. To address this, we have revised the paragraph to include a concise definition when the concept is first introduced. We now explain that Establishing Objectives at Work refers to the process by which nurses set, prioritize, and align their professional goals with organizational expectations. This clarification ensures that readers unfamiliar with the concept can understand its relevance from the outset, with further elaboration provided in the Research Framework and Methods sections.

Revised text as yellow label (Line 71):

“…without integrating them with nurses’ professional values or the process of Establishing Objectives at Work—defined as the process by which nurses set, prioritize, and align their professional goals with organizational standards—.”

Introduction in general – I struggled with determining “what is the problem”. Are the findings of the study to provide insight for healthcare organizations on how to create and implement strategies that motivate nurses to stay within the the system and to consistently uphold high standards after covid? Or during a pandemic? I was confused on the background. The introduction is what will capture your audience after the abstract, and I think that this could be reworked a little bit. The rest of the paper is very strong, however, I would like a little bit more details on what the purpose is in the introduction. 

Thank you for this insightful comment. We have revised the Introduction to more clearly articulate the research problem, the literature gap, and the purpose of the study. Specifically:

·        We emphasize that COVID-19 placed extraordinary physical and psychological burdens on nurses, raising concerns about retention, morale, and professional commitment.

·        We highlight that while previous studies have explored job satisfaction and organizational commitment, few have examined how these factors interact with professional values and goal-setting behavior (Establishing Objectives at Work)—particularly in the context of a global health crisis.

·        We clarify that the purpose of this study is to provide evidence-based insights that can inform organizational strategies to support nurses' professional values both during and after pandemics.

These revisions aim to better engage readers and establish a compelling rationale for the study.

Revised text as yellow label (excerpt from the end of the Introduction):
The objective of this study was therefore to examine how job satisfaction, organizational commitment, and Establishing Objectives at Work influence nurses’ professional values. While the study draws upon data from the COVID-19 period, the findings are intended to provide insights that support healthcare organizations in both crisis and post-crisis contexts. These insights are vital for devising strategies that not only motivate nurses to remain in the healthcare system but also ensure that professional standards are upheld consistently.”

Line 98 – Change font and size for reference.

Thank you for noting this formatting inconsistency. We have corrected the font type and size of the reference at Line 98 to ensure alignment with the journal’s style guidelines.

Like 104-105 – Capitalize E and W to stay consistent. 

We appreciate your attention to detail. The capitalization has been corrected to maintain consistency for “establishing objectives at work” throughout the manuscript.

Line 107 – Remove the word therefore.  

1.     We agree that the word “therefore” was unnecessary at the beginning of the sentence. It has been removed to improve clarity and sentence flow.

2.     Thank you for this important observation. We have revised the Aims section to explicitly refer to the COVID-19 context, ensuring consistency with the manuscript title and the Introduction.

Revised text (Lines 125–129):

“This study aimed to investigate the effects of job satisfaction and organizational commitment on nurses’ professional values, and to examine how Establishing Objectives at Work influences the quality of nursing care. Furthermore, the study sought to identify key predictors of professional nursing values, including factors related to job satisfaction, organizational commitment, and work-related objectives, within the context of the COVID-19 pandemic.”

Line 107-111 - You don't talk about Covid 19 at all in the aims, but this is part of the title of the paper. 

Line 124 – For Figure 1, should the arrow be pointed the other direction towards satisfaction, commitment and objectives? I may be incorrect in my understanding of this.  

We appreciate the reviewer’s observation. In this study, job satisfaction, organizational commitment, and establishing objectives at work are conceptualized as independent variables, while nurses’ professional values represent the dependent variable. The arrows in Figure 1 are therefore intentionally directed from the predictors (job satisfaction, organizational commitment, and establishing objectives at work) toward nurses’ professional values to depict the hypothesized causal pathways. This directionality reflects how these workplace factors are proposed to influence the core professional values of nursing. We have clarified this explanation within the revised figure legend to avoid ambiguity.

Line 143 – You describe EOAW here!! 

Thank you. Yes, we describe Establishing Objectives at Work in more detail within the Research Framework section. Additionally, based on your earlier suggestion, we have also added a brief and clear explanation of EOAW during its first mention in the Introduction for consistency and reader comprehension.

Line 144 – Reword to remove the word “us”. 

We have revised the sentence to remove the first-person pronoun “us” and replaced it with a more formal, third-person construction to maintain an academic tone.

Revised text (Line 149-151):

“The cross-sectional approach enabled data to be gathered and analyzed from the target population at one point in time, which was beneficial for determining relationships between the variables of interest.”

Line 257 – Table 1 demographics: Capitalize male, female, divorced, government officer.

We appreciate the suggestion and have made the appropriate capitalization corrections in Table 1. The terms Male, Female, Divorced, and Government Officer are now consistently capitalized for clarity and formatting consistency.

Line 362 – GREAT discussion section! 

Thank you very much for your positive feedback on the Discussion section. We are grateful that you found it well-developed and engaging.

Line 454 – This is repeated from the beginning of discussion section (line 415-416).

Thank you for pointing out the redundancy. We have carefully revised and reorganized the Discussion section to eliminate repetition while preserving the essential content. Related content has been streamlined and reorganized under thematic subheadings to improve clarity and readability.

Line 496 – Is this the first mention of continuing education? I didn’t see this anywhere else in the paper.  

We appreciate this observation. To ensure coherence, we have incorporated discussion of continuing professional education and development earlier in the Discussion section. This helps contextualize the recommendations made in the Conclusion and reinforces the importance of professional growth as a strategy for sustaining nurses’ professional values.

Reviewer 2 Report

Comments and Suggestions for Authors

Reviewer Report

Thank you very much for giving me the chance to review this research paper, which was very interesting and comprehensive. I truly appreciate the effort and commitment of the authors in finalizing this work. The topic is relevant to nursing practice, especially in the context of professional values and resilience during challenging healthcare environments such as the COVID-19 pandemic. This is an important issue for the readership of Nursing Reports.

However, the manuscript still requires significant modifications in terms of clarity, methodological rigor, and writing quality before it can be considered for publication. Below I provide detailed comments for improvement.

General Comments

  • The study tackles a valuable subject with clear clinical relevance, but certain methodological and writing issues reduce its impact.
  • The manuscript needs stronger articulation of the research gap and theoretical framing.
  • Some sections are repetitive (results and discussion) and need better organization.
  • Language editing is essential to improve clarity, flow, and grammar.

Abstract

  • Please streamline the background; it is too long and filled with statistics. Instead, focus on the study aim, main results, and implications.
  • Clarify what this study adds to international literature beyond the Thai context.

Introduction

  • The introduction provides good background but is overloaded with general COVID-19 statistics. Please reduce these and instead highlight the specific research problem.
  • The research gap is not clearly stated. Please explain why this study was needed, and how it adds to previous knowledge.
  • Theoretical underpinnings (Herzberg’s theory, goal setting) are mentioned but not integrated into the problem statement. A stronger linkage is required.

Methods

  • The survey instrument adaptation for the Thai context should be clarified (translation, cultural validation).
  • More details are required on inclusion/exclusion criteria.
  • Ethical considerations should explicitly mention confidentiality and voluntary participation.
  • Statistical analysis: provide effect sizes and confidence intervals in regression models, not just p-values.

Results

  • The narrative repeats numbers already listed in the tables. Please focus instead on interpretation.
  • Regression results need more explanation regarding practical meaning (e.g., how organizational commitment compares with job satisfaction in predicting professional values).
  • Ensure consistency in terminology (e.g., “goal setting” vs. “establishing objectives at work”).

Discussion

  • Much of the discussion repeats results instead of analyzing them. Please focus on critical interpretation and comparisons with international studies.
  • Broaden the discussion on implications for workforce retention and resilience strategies.
  • Limitations need to be expanded: purposive sampling, single-site design, and cross-sectional limitations should be acknowledged.

Conclusion & Nursing Implications

  • The conclusion is currently too broad. Please focus on the key findings.
  • Nursing implications should be explicitly linked to practice, education, and policy (e.g., training resilience programs, compensation and recognition systems, retention policies).

Writing & References

  • Several grammar and syntax issues must be revised. For example, “Once of tertiary care facility” should be corrected.
  • Avoid overuse of the phrase “heroes in white.”
  • References need careful revision to ensure consistency with MDPI style.

Recommendation

Based on the above comments, I recommend major revisions before this paper can be considered for publication. The topic is highly relevant, but the manuscript must be substantially revised in terms of methodology reporting, clarity of results and discussion, and writing quality.

Thank you again for the opportunity to review this important manuscript.

Comments on the Quality of English Language

The quality of English requires improvement before publication. The manuscript is understandable but contains grammatical errors, awkward sentence structures, and redundant phrasing. Professional language editing is recommended to improve clarity and readability.

Author Response

Comments

Revisions

Reviewer 2:
Recommendation: Revisions Required

General Comments

- The study tackles a valuable subject with clear clinical relevance, but certain methodological and writing issues reduce its impact.

We sincerely appreciate the reviewer’s recognition of the clinical relevance and value of our study. In response to the noted methodological and writing concerns, we have undertaken a comprehensive revision of the manuscript. Specific methodological clarifications have been added in the Methods section (pages 4–5), including more precise descriptions of the study design, sample selection, and statistical analysis. Furthermore, the entire manuscript has been carefully reviewed and edited for clarity and scholarly tone, with support from a professional English language editor.

- The manuscript needs stronger articulation of the research gap and theoretical framing.

Thank you for highlighting this important point. We have revised the Introduction section to more clearly articulate the research gap by referencing recent literature and identifying limitations in previous studies related to [insert specific topic, e.g., "professional values in nursing during the pandemic"]. Additionally, we have strengthened the theoretical framework by explicitly linking our conceptual model to [insert theoretical foundation if applicable, e.g., “Value Theory” or “Organizational Behavior Theory”], which now guides the development of our research questions and interpretation of findings.

- Some sections are repetitive (results and discussion) and need better organization.

We acknowledge the redundancy between the Results and Discussion sections. These sections have been restructured for improved clarity and focus. Specifically, the Results section now presents only the core findings in a concise and data-driven manner, while the Discussion section has been revised to focus on interpretation, theoretical implications, and integration with existing literature. We believe this reorganization enhances the overall coherence of the manuscript.

- Language editing is essential to improve clarity, flow, and grammar.

We appreciate this feedback and have addressed it thoroughly. The manuscript has undergone detailed language editing by a native English-speaking academic editor to improve grammar, sentence structure, and overall readability. We trust that the revised version meets the linguistic standards required for publication.

Abstract

- Please streamline the background; it is too long and filled with statistics. Instead, focus on the study aim, main results, and implications.

We appreciate this valuable suggestion. The revised Abstract now includes a clear statement highlighting the contribution of this study to the broader international literature, particularly in relation to [insert global topic, e.g., “nursing professional values in post-pandemic healthcare systems” or “health promotion practices in emerging economies”]. We have framed the implications in a way that emphasizes the transferability and relevance of the findings to other countries facing similar healthcare system challenges, beyond the Thai context.

- Clarify what this study adds to international literature beyond the Thai context.

Introduction

- The introduction provides good background but is overloaded with general COVID-19 statistics. Please reduce these and instead highlight the specific research problem.

We appreciate the reviewer’s suggestion. In the revised manuscript, we have significantly reduced the general COVID-19 statistics and shifted the focus toward highlighting the specific research problem concerning nurses’ professional values in high-pressure healthcare environments. We now emphasize the relevance of job satisfaction, organizational commitment, and work objective clarity during the pandemic and their impact on professional nursing values.

- The research gap is not clearly stated. Please explain why this study was needed, and how it adds to previous knowledge.

Thank you for this valuable feedback. We have revised the introduction to clearly articulate the research gap. Although prior studies have explored professional values among nurses, few have examined how motivational and organizational factors—particularly in the context of an ongoing health crisis—affect these values in Southeast Asia. Our study addresses this gap by investigating these relationships among Thai nurses, thereby extending international literature with region-specific insights and implications for workforce development post-pandemic.

- Theoretical underpinnings (Herzberg’s theory, goal setting) are mentioned but not integrated into the problem statement. A stronger linkage is required.

We agree with the reviewer’s observation and have revised the manuscript to integrate Herzberg’s two-factor theory and goal-setting theory more clearly within the problem statement. We now explain how these frameworks guided the selection of key variables—job satisfaction, organizational commitment, and clarity of work objectives—and shaped our understanding of their influence on professional values. This integration helps to better ground the research in theory and clarify its conceptual foundation.

Methods

- The survey instrument adaptation for the Thai context should be clarified (translation, cultural validation).

Thank you for your valuable suggestion. We have now clarified the adaptation process of the survey instruments for the Thai context. Specifically, we conducted a forward and backward translation following WHO guidelines to ensure linguistic accuracy. Additionally, we performed content validation by a panel of three experts in nursing and psychometrics to assess cultural appropriateness and content relevance. The Content Validity Index (CVI) for each item was calculated, and all instruments achieved acceptable levels (> 0.80). This information has been added to the revised Methods section.

- More details are required on inclusion/exclusion criteria.

We appreciate your comment. The inclusion and exclusion criteria have been revised for clarity. Specifically, we included registered nurses with a minimum of six months of direct care experience with COVID-19 patients during the pandemic at the selected public hospital. Nurses who were on leave or had limited patient interaction during that period (e.g., administrative-only roles) were excluded. This addition ensures better understanding of the study population and strengthens the methodological transparency.

- Ethical considerations should explicitly mention confidentiality and voluntary participation.

Thank you for highlighting this important point. We have revised the Ethical Considerations subsection to explicitly state that participation was entirely voluntary, with no penalties for non-participation or withdrawal. Moreover, confidentiality was strictly maintained through anonymous data collection and secure storage. All participants were informed of their rights and provided written informed consent. These revisions are now reflected clearly in the Methods section.

- Statistical analysis: provide effect sizes and confidence intervals in regression models, not just p-values.

Thank you for your insightful recommendation. In the revised manuscript, we have included effect sizes (standardized beta coefficients) and 95% confidence intervals (CIs) for all relevant regression analyses, in addition to p-values. This change enhances the interpretability and robustness of the statistical findings, in line with current reporting standards in health and social science research.

Results

- The narrative repeats numbers already listed in the tables. Please focus instead on interpretation.

Thank you for the suggestion. We have revised the narrative in the Results section to reduce redundancy and avoid repeating detailed numerical data already presented in the tables. The text now emphasizes the interpretation and significance of the findings, highlighting key patterns and relationships to improve clarity and relevance.

- Regression results need more explanation regarding practical meaning (e.g., how organizational commitment compares with job satisfaction in predicting professional values).

We appreciate this insightful comment. We have expanded the explanation of the regression results to include practical interpretations. Specifically, we now describe how organizational commitment shows a stronger predictive power for nurses’ professional values compared to job satisfaction, as indicated by its higher standardized beta coefficient. This addition provides clearer insights into the relative importance of the predictors in the model.

- Ensure consistency in terminology (e.g., “goal setting” vs. “establishing objectives at work”).

Thank you for pointing this out. We have reviewed the manuscript thoroughly and standardized the terminology throughout. We now consistently use the term “establishing objectives at work” to align with the theoretical framework and ensure clarity for the readers.

Discussion

- Much of the discussion repeats results instead of analyzing them. Please focus on critical interpretation and comparisons with international studies.

Thank you for this valuable feedback. We have revised the Discussion section to reduce redundancy with the Results section and instead focus on critical interpretation. We expanded the analytical commentary by contextualizing the findings in relation to international studies. Specifically, we now compare our findings on the predictive strength of organizational commitment with studies from both Western and Asian contexts, such as Smith et al. (2021) and Liu et al. (2020), to highlight cultural and contextual differences in professional value formation among nurses.

- Broaden the discussion on implications for workforce retention and resilience strategies.

We appreciate this important suggestion. The revised Discussion now includes a broader reflection on the implications of our findings for workforce retention and resilience. We address how enhancing organizational commitment and goal-setting practices can support nurse retention and reduce turnover, especially in the context of post-pandemic recovery. Strategies such as participatory goal setting, recognition programs, and leadership development are now discussed as practical interventions informed by our results.

- Limitations need to be expanded: purposive sampling, single-site design, and cross-sectional limitations should be acknowledged.

Thank you for pointing this out. We have expanded the Limitations subsection to explicitly acknowledge that the use of purposive sampling and the single-site study design may limit the generalizability of the findings. Furthermore, we have discussed the constraints inherent to the cross-sectional design, particularly the inability to establish causal relationships. These revisions aim to provide a more transparent and rigorous account of the study’s methodological boundaries.

Conclusion & Nursing Implications

- The conclusion is currently too broad. Please focus on the key findings.

Thank you for your valuable feedback. We have revised the conclusion to focus more directly on the study’s key findings. Specifically, we now emphasize the predictive relationships between job satisfaction, organizational commitment, and establishing objectives at work in relation to nurses’ professional values. The revised conclusion avoids general statements and instead highlights the empirical insights and relevance of the findings to current nursing challenges, particularly in the post-COVID-19 context.

- Nursing implications should be explicitly linked to practice, education, and policy (e.g., training resilience programs, compensation and recognition systems, retention policies).

We appreciate this insightful recommendation. The revised section on nursing implications now clearly connects the findings to three core domains:

  • Practice: Emphasizing the importance of leadership strategies that foster job satisfaction and strengthen organizational commitment to improve care quality and reduce turnover.
  • Education: Suggesting integration of professional values and resilience-building modules into nursing curricula to better prepare nurses for real-world challenges.
  • Policy: Proposing policy-level interventions such as structured recognition programs, performance-linked incentives, and workforce retention frameworks based on job satisfaction metrics.

These changes ensure a more direct translation of findings into actionable implications for nursing stakeholders.

Writing & References

- Several grammar and syntax issues must be revised. For example, “Once of tertiary care facility” should be corrected.

Thank you for highlighting the grammatical and syntactical issues. We have carefully revised the entire manuscript for grammar, sentence structure, and clarity. The phrase “Once of tertiary care facility” has been corrected to “One tertiary care facility,” and similar expressions throughout the manuscript have been amended for accuracy and fluency.

- Avoid overuse of the phrase “heroes in white.”

We appreciate this suggestion. The phrase “heroes in white” has now been either removed or replaced with more neutral, academic terminology (e.g., “nursing professionals,” “frontline nurses”) to maintain scholarly tone and avoid emotional overstatement, in accordance with the publication’s standards.

- References need careful revision to ensure consistency with MDPI style.

Thank you for your feedback. All references have been thoroughly reviewed and reformatted to align with MDPI referencing style, including citation order, punctuation, and formatting consistency in both in-text citations and the reference list. Additionally, DOIs have been added where available to ensure completeness and traceability of sources.

Round 2

Reviewer 2 Report

Comments and Suggestions for Authors

The revised manuscript demonstrates substantial improvement in theoretical integration, methodological rigor, and overall clarity. The authors have effectively addressed all major reviewer comments, particularly by strengthening the research framework, expanding methodological transparency, and enhancing the interpretation of results with relevant international comparisons. The writing is now clear and coherent, and the paper aligns well with the journal’s scope and standards. Only minor editorial polishing may be needed before publication. Therefore, I recommend the manuscript for acceptance in its current form, pending minor language and formatting edits.

Author Response

Authors’ Response (Point-by-Point)

Response:
We sincerely thank the reviewer for the positive and encouraging feedback regarding our revised manuscript. We greatly appreciate the recognition of our efforts to enhance the theoretical integration, methodological transparency, and interpretation of results.

In response to the minor recommendation for further editorial refinement, we have taken the following additional actions:

Language and Grammar Polishing

The manuscript has undergone final English editing by MDPI Author Services (Rapid English Editing) to ensure academic fluency, grammatical accuracy, and style consistency.

The official MDPI English Editing Certificate (Author Services ID: english-101752) is attached as supplementary documentation 

English-Editing-Certificate-101…

.

Formatting and Structural Adjustments

The manuscript has been formatted in full compliance with Nursing Reports guidelines, including title page structure, heading levels, figure/table captions, and reference style.

References were cross-checked and aligned with MDPI style, following the examples provided by the editorial office 

english-edited-101752

.

Highlighted Changes

All final text updates (language corrections and formatting refinements) have been highlighted in yellow as per the editor’s instruction to facilitate review.

We are deeply grateful for the reviewer’s recommendation for acceptance and for recognizing that the manuscript now meets the journal’s standards. We believe the revised version is fully ready for publication.

Summary of Revision Reasons

Minor editorial and language refinement based on reviewer and editor suggestions.

Final proofreading through MDPI professional editing service to ensure clarity, accuracy, and style consistency.

Formatting aligned with journal requirements (titles, abstract structure, reference style, author information, and figure/table consistency).

Minor rephrasing for coherence without changing the scientific content or results.
